# I Am Conscious, Therefore, I Am: Imagery, Affect, Action, and a General Theory of Behavior

**DOI:** 10.3390/brainsci9050107

**Published:** 2019-05-10

**Authors:** David F. Marks

**Affiliations:** Independent Researcher, Arles, Bouches-du-Rhône, 13200 Provence-Alpes-Côte d’Azur, France; editorjhp@gmail.com

**Keywords:** vividness, mental imagery, consciousness, cognitive neuroscience, neuroimaging, cognitive psychology, behavior, verbal report, phenomenology, perception

## Abstract

Organisms are adapted to each other and the environment because there is an inbuilt striving toward security, stability, and equilibrium. A General Theory of Behavior connects imagery, affect, and action with the central executive system we call consciousness, a direct emergent property of cerebral activity. The General Theory is founded on the assumption that the primary motivation of all of consciousness and intentional behavior is psychological homeostasis. Psychological homeostasis is as important to the organization of mind and behavior as physiological homeostasis is to the organization of bodily systems. Consciousness processes quasi-perceptual images independently of the input to the retina and sensorium. Consciousness is the “I am” control center for integration and regulation of (my) thoughts, (my) feelings, and (my) actions with (my) conscious mental imagery as foundation stones. The fundamental, universal conscious desire for psychological homeostasis benefits from the degree of vividness of inner imagery. Imagery vividness, a combination of clarity and liveliness, is beneficial to imagining, remembering, thinking, predicting, planning, and acting. Assessment of vividness using introspective report is validated by objective means such as functional magnetic resonance imaging (fMRI). A significant body of work shows that vividness of visual imagery is determined by the similarity of neural responses in imagery to those occurring in perception of actual objects and performance of activities. I am conscious; therefore, I am.

## 1. Preliminaries

A General Theory of Behavior concerns the “I am” control center for (my) thoughts, (my) feelings, and (my) actions with (my) conscious mental imagery. Thus, 382 years after Descartes stated his “cogito, ergo sum” discovery, the evidence from cognitive neuroscience suggests the possibility of a more satisfying claim: “I am conscious, therefore, I am”. The certainty of human existence relies not on the ability to think, but on the richer endowment provided by consciousness. It is not too large a stretch to assert that consciousness brought humans their privilege of pole position in the food chain. Without consciousness, humans would not benefit from what, I claim, is the pre-eminent force of nature that drives evolution, homeostasis. Organisms are adapted to each other and the environment as conscious beings because they possess an inbuilt striving toward stability, security, and equilibrium. Arguably, human homeostatic activity in niche construction generates less variation in the source of selection than where there is no feedback from organisms’ activities to the environment [1].

The General Theory connects imagery, affect, and action with consciousness and the primary motivation that is psychological homeostasis. It is suggested that homeostasis provides a unifying concept across biology and psychology. Psychological homeostasis is as important to the organization of mind and behavior as physiological homeostasis to the organization of bodily systems. According to the theory, psychological homeostasis is available to organisms with consciousness, a process that embraces a near infinite supply of quasi-perceptual-affective images independently of the retina and sensorium. Homeostatic striving for security, stability, and equilibrium is a precondition for well-being. One major form of behavioral homeostasis is niche construction which alters ecological processes, modifies natural selection, and contributes to inheritance through ecological legacies. To do full justice to this very broad topic within the confines of the special issue, this invited review article refers to previous publications as supporting material, with all of the detailed references, a more complete explanation of the ideas, and the ongoing state of the research. One major purpose of the review is to present a summary of the evidence that mental imagery plays an essential role in the control of behavior. Mental imagery is an essential component of a new General Theory of Behavior involving homeostasis. A second purpose of this review is to explore the executive function of consciousness in the organization and control of behavior.

Central to the General Theory is the construct of mental image vividness, that combination of clarity and liveliness, so beneficial to consciousness and all its works: imagining, remembering, thinking, feeling, predicting, planning, pretending, dreaming, and acting. Introspective reports and objective indicators suggest the existence of wide individual differences in mental imagery vividness. Only in rare cases do is there evidence of absolutely no vividness at all. Vividness was studied under controlled conditions with standardized questionnaires, including the Vividness of Visual Imagery Questionnaire (VVIQ) [2,3,4]. An alternative method for studying vividness in experimental settings is to ask participants to provide vividness ratings (VR) on a “trial-by-trial” basis corresponding to the subjective experience at each particular moment in time [5]. This procedure avoids the problem leveled at the VVIQ that people cannot evaluate their private imagery along a common vividness scale because they have no objective reference points. However, this objection is refuted by validating data from controlled experiments. Assessment of vividness using introspective report was validated in multiple studies like those reviewed below using objective measures such as functional magnetic resonance imaging (fMRI). Also, the two end-points of the vividness scale are universally anchored by the descriptor “perfectly clear and as vivid as normal vision” at one end and “no image at all, you only ‘know’ that you are thinking of an object” at the other end of the scale.

Philosophers and psychologists discussing the vividness construct and its measurement sometimes displayed the characteristics of a “streetlight effect”, the observational bias that occurs when searching for something and looking only where it is easiest, where there is light [6]. Searching in familiar terrains of logic and theory of mind while ignoring less familiar terrains of neuroscience and psychology can produce a fragmentary review and false conclusions. These problems come to the fore in a recent paper on “imaginative vividness” by Kind [7], in which the author suggests that it would be “best to retire our reliance on this notion entirely”. These dismissive conclusions, I suggest, are based on incomplete examination of the evidence and faulty analysis of vividness and its phenomenology. For pragmatic reasons, I confine the present discussion to visual imagery, although similar principles are thought to apply to images of all modalities.

The General Theory assumes there is universal and constant striving for stability and equilibrium, a process termed “psychological homeostasis” [1]. Neuroscientific theories automatically fall under suspicion of material reductionism. However, a reductive approach is not required or desired and is not the approach taken here. The idea that behavior is reducible to physico-chemical reactions or to mechanistic “cogs and wheels” is rejected. This point is stated in the following working principle:
**Working** **Principle:*****The voluntary behavior of conscious organisms is guided by a universal striving for equilibrium, a striving with purpose, desire, and intentionality.***

Psychological homeostasis, as a process of consciousness, is intentional, purposeful, and driven by the desire for security, safety, and equilibrium. It is necessary to assume that the mind/body system as a whole can be studied using objective methods. For one hundred years, mental imagery, vividness, and consciousness remained under-investigated in psychology and neuroscience for being too “subjective”. The so-called “subjective” processes of vividness, mental imagery, and consciousness are amenable to objective study with tools and methods designed specifically for the purpose. This article indicates how these once “tabooed” areas became part of the scientific mainstream. A primary function of consciousness is the mental rehearsal of adaptive, goal-directed action through the experimental manipulation of perceptual-motor imagery. Every cycle of mental activity includes a goal, the means to reach it, and the consequences for the organism and objective world. The control system has a “meta level”, a “schema level”, and an “automatized level” in a hierarchical relationship. The meta level, which is a major function within “consciousness”, sets the goals for a project of activity. The schema level monitors and controls the flow of action, making moment-by-moment adjustments in light of the goal set for it by the meta level above. The automatized level carries out repetitive and routine, everyday tasks that require zero planning or attention from higher up. When one hammers a nail into a piece of wood, one checks if the nail straight, if it is going in properly, whether it hit an obstacle, and so on. In cooking a soup, one tastes and seasons it with salt and pepper, not too much and not too little. In driving on the highway, one keeps to one’s lane, attends to the road, monitors the other vehicles, and keeps to the speed limit, constantly vigilant for signs and warning signals, adjusting, adapting, and correcting. Conscious goal-setting lends purpose to our every action, striving for equilibrium with desire and intentionality.

## 2. Vividness

What exactly do I mean by “vividness”? It is necessary to be clear to avoid misunderstanding. I need search no further than the definition provided in an earlier publication: “a combination of clarity and liveliness. The more vivid an image, therefore, the closer it approximates an actual percept” [4]. The two elements, clarity and liveliness, are equally important. From the get-go, it is noted that vividness is independent of detail: there can be a lot of detail or none. Clarity and liveliness are the defining criteria. A vivid mental image of a red flag blowing in the wind requires nothing more than color, movement, and clarity—the quality of being clear, distinct, and intentional—a red flag, not a red rag. Liveliness is vitality, being animated, and the ability to evoke feelings of, say, attraction or repulsion. A vivid image is alive with vitality, energy, ebullience, and the potential to evoke activity and feeling.

A “life-like” image allows a person to experience it in a very real way. “Evoke” means to cause someone to sense or feel something. I appreciate this fact from personal experience. When I am watching a film of people lighting a camp fire, or smoking a cigarette, I actually can “smell” the smoke at the camp fire or the stench of a cigarette. These olfactory imaginings occur as if I am actually carrying out the activity, albeit on a lessor scale. Note that it is an activity, not a picture. Evoking a vivid image produces a life-like activity of “seeing”, “hearing”, “tasting”, “smelling”, “touching” or “feeling” something; mental imagery is a sensory-affective process that resembles, but is not identical to, perception with action.

Imagery, observation, and activity are produced by similar neural processes within a single system of representation in the brain. The evidence shows that the neurophysiological mechanisms that are active during physical skill acquisition are active during imagery and observation of the same skill [8]. Visual ideas may be cashed out as actions or they may be entirely covert. There is a ceaseless progression of ideas and associations in consciousness with or without the path markers of vivid imagery. In truth, every person has an Angie Thomas or a Khaled Hosseini sitting inside. Without vividness, however, no *The Hate U Give* or *Kite Runner*. There would also be fewer scientific discoveries—no Maxwell’s demon, Einstein’s elevator, or Schrödinger’s cat (Figure 1).

Whatever else humans can do, our visual imagery ability is vital to our humanity, whether it is the everyday problem-solving required for stability, safety, and survival at home, in the workplace, or in the community. The essentially visual nature of thinking is “reflected” in the words used in conversation about “seeing”, “views”, “standpoints”, “outlooks”, “perspectives”, “prospects”, “angles”, and “horizons”, to mention only a small sample. Antonio Damasio explains the value of mental imagery to “creative intelligence” in human evolution: “Creative intelligence was the means by which mental images and behaviors were intentionally combined to provide novel solutions for the problems that humans diagnosed and to construct new worlds for the opportunities humans envisioned” [9] (p. 71).

The analysis of vividness risks entering a cul-de-sac when entertaining unhelpful metaphors such as the Platonic “picture” theory or the “descriptive”, propositional theory [10] are adduced as if our mental hardware is akin to the technology for editing digital photographs or code. Mental images are not internal “pictures” or “photographs”; they are not anything like them. The mistake of the picture theory created insuperable problems and confusion. This review offers a different perspective on the nature and function of mental imagery, beginning with the measurement of vividness. It was demonstrated that conscious imagery is not equally vivid in all people and the reasons for, and consequences of, this fact stimulated a great deal of research. In the next section, I review one of the instruments employed in this research.

## 3. Vividness of Visual Imagery Questionnaire

To date, around 2000 studies have used the VVIQ or Vividness of Movement Imagery Questionnaire (VMIQ) [11] as a measure of imagery vividness. In this article, I address the VVIQ and leave the VMIQ to another occasion. The VVIQ is a self-report measure of the clarity and liveliness of visual imagery and, in so doing, aims to evoke images that vary in vividness, ambiance, and feeling as well. The instructions state the following:

“Visual imagery refers to the ability to visualize, that is, the ability to form mental pictures, or to ‘see in the mind’s eye’. Marked individual differences are found in the strength and clarity of reported visual imagery and these differences are of considerable psychological interest.

The aim of this test is to determine the vividness of your visual imagery. The items of the test will possibly bring certain images to your mind. You are asked to rate the vividness of each image by reference to the five-point scale given below. For example, if your image is ‘vague and dim’, then give it a rating of 4. After each item, write the appropriate number in the box provided. The first box is for an image obtained with your eyes open and the second box is for an image obtained with your eyes closed. Before you turn to the items on the next page, familiarize yourself with the different categories on the rating scale. Throughout the test, refer to the rating scale when judging the vividness of each image. Try to do each item separately, independent of how you may have done other items.

Complete all items for images obtained with the eyes open and then return to the beginning of the questionnaire and rate the image obtained for each item with your eyes closed. Try and give your ‘eyes closed’ rating independently of the ‘eyes open’ rating. The two ratings for a given item may not in all cases be the same.” [4].

The five-point rating scale of the VVIQ is presented in Table 1. Some researchers prefer to reverse the numerical scale to make 5 = perfectly clear and as vivid as normal vision, and 1 = no image at all, you only “know” that you are thinking of an object.

The 16 items are arranged in blocks of four, in which each has a theme and at least one item in each cluster describes a visual image that includes movement (Table 2). Each theme provides a narrative to guide a progression of mental imagery. It is noted that at least one item in each cluster describes an activity or movement, indexing liveliness. The aim of the VVIQ is to assess visual imagery vividness under conditions which allow a progressive development of scenes, situations, or events as naturally as possible. The items are intended to evoke sufficient interest, meaning, and affect conducive to image generation. Participants rate the vividness of their images separately with eyes open and eyes closed.

For a small minority of people, the capacity for visual imagery is unavailable. In the absence of mental imagery, consciousness consists of “unheard” words, “unheard” music, and “invisible” imagery. This minority needs to employ more generic, verbal methods to recall events, and to plan goals and future activity—compensatory strengths that remain under-investigated.

## 4. The Nature and Function of Imagery

A large body of psychological and neuroscientific research is consistent with the tenet that imagery is functionally equivalent to, but not identical to, perception. Research summarized in a later section indicates similar anatomical patterning of neural activity in the cerebral cortex. I am discussing a graded phenomenon in which “the ordinary course of thought involves an interaction at sensory input with the central processes…” [12] (p. 476), which can range from the very vivid to the completely abstract. As in perception, potentiality for action and a corresponding degree of anticipatory feeling and volition are part and parcel of a vivid mental imagery experience. Mental images come laden with associations in the form of feelings, e.g., attraction, calm, tranquility, fear, anger, or anticipation, useful “reflections” on life events in psychotherapy through image evocation in sessions of imagery therapy [13]. It is these attributes of mental imagery that enable works of literature to move readers to experience narrative and characters as if they are “real”. Yet, it is vividness itself rather than arousal that is most closely correlated with the aesthetic appreciation of poetry, such as haiku or sonnets [14].

Another method to explore a person’s imagery experience is to provide simple color suggestions while the participant looks into the center of a circle drawn on paper—the Open Circle Test [15]. When a vivid imager is invited to describe the imagery evoked by a color name, then a sequence of lively images may be experienced including isomorphisms and visual metaphors. For example, in one 38-year-old woman, the suggestion “yellow” produced a sequence starting with a yellow canary and transitioning into a variety of increasingly complex, dynamic images (Figure 2).

Each image connected and overlapped with the previous one. The sequence could have been continued indefinitely but we stopped after four transitions. The dynamic images triggered by a simple color name show the defining feature of liveliness. This attribute is neglected in practically all analyses of imagery. Vividness, action, and affect are the foundation stones for the General Theory of Behavior and of consciousness itself.

An extensive literature on mental “practice” refers to both imagery “rehearsal” and mental “simulation” [16,17]. Imagery is routinely and systematically employed in preparation and rehearsal of sports activity and was shown to produce enhanced performance across a wide variety of skillsets [18,19]. Studies of skilled performers show that activity cycles are more effectively rehearsed when they incorporate vivid imagery [20]. Studies of Olympic athletes and performers capable of specialist skills suggest that high imagery vividness is of most benefit to performances that have significant perceptual-motor components or require visualization of complex interactions at the object level [21].

Converging evidence suggests that mental simulation of movement and actual movement share similar neurocognitive and learning processes, leading to considerable interest in imagery simulation of movement as a therapeutic tool in the rehabilitation of stroke patients, patients with Parkinson’s disease, and other neurological syndromes [22]. Conscious imagery enables the user to explore, select, and prepare physical and social activity. However, mental simulation is not a process of inspecting a holistic visual image like a picture or photograph in the “mind’s eye”. Mental simulations are constructed piecemeal, include non-visible properties, and can be used in conjunction with non-imagery processes, such as task decomposition and rule-based reasoning [23].

A common neural basis exists for imitation, observational learning, and motor imagery. During mental simulation, the excitatory motor output generated for executing the action is inhibited. The autonomic system is also activated during motor imagery. The principal function of consciousness is to plan and make predictions about the consequences of actions. Simulation enables the imager to mentally try out a sequence of goals, schemata, and actions that minimize hazard, loss, and pain.

The principal measures of vividness, the VVIQ and VR, are strongly associated with performance in different kinds of tasks: self-report, physiological motor, perceptual, cognitive, and memory [4,5,24,25]. To quote Runge et al. [5], “vividness can be considered a chief phenomenological feature of primary sensory consciousness, and it supports the idea that consciousness is a graded phenomenon”. Recent research reviewed below showed that reported vividness is associated with early visual cortex activity relative to the whole-brain activity measured by functional magnetic resonance imaging (fMRI) and the performance on a novel psychophysical task.

Vividness of visual imagery correlates with fMRI activity in early visual cortex scores, demonstrating that higher visual cortex activity indexes more vivid imagery. Variations in imagery vividness depend on a large network of brain areas, including frontal, parietal, and visual areas. The more similar the neural response during imagery is to the neural response during perception, the more vivid or perception-like the imagery experience will be. From these findings, it can be concluded that an image is an idea with visual attributes. The more vivid the image, the more strongly a person will be aware of it. Upon reflection of the alternative actions available, it is possible to inhibit certain actions and implement others, or to keep actions “on hold” for the future. Thus, consciousness is able to facilitate successful striving toward goals, including the construction of new niches and environments, and thereby the effectiveness of type II homeostasis, providing a significant evolutionary advantage through niche construction and other adaptive mechanisms [1].

The evidence on the VVIQ [5,25] permits the following conclusions:The VVIQ is only minimally contaminated or not contaminated at all by extraneous variables such as social desirability.The VVIQ has acceptable levels of split-half and test-retest reliability.The VVIQ has acceptable validity coefficients with other verbal-report measures of imagery and with physiological, motor, memory, and cognitive tasks.The VVIQ has excellent criterion validity coefficients with perceptual tasks, exceeding those obtained with other self-report measures of imagery.The ability to produce vivid visual imagery is of most benefit in tasks that are heavily loaded with perceptual-motor components and is of considerably less benefit to memory tasks.A recent meta-analysis reported large effect size estimates (ESEs) for both VR and VVIQ measures, with larger ESEs for neuroscientific than behavioral-cognitive measures [5].

I summarize here two key findings.

*STUDY 1:**Vividness of mental imagery: individual variability can be measured objectively* [26]. “When asked to imagine a visual scene, such as an ant crawling on a checkered table cloth toward a jar of jelly, individuals subjectively report different vividness in their mental visualization. We show that reported vividness can be correlated with two objective measures: the early visual cortex activity relative to the whole-brain activity measured by functional magnetic resonance imaging (fMRI) and the performance on a novel psychophysical task. These results show that individual differences in the vividness of mental imagery are quantifiable even in the absence of subjective report” [26] (p. 474).

“Results 3.1. Vividness of visual imagery correlates with fMRI activity in early visual cortex scores. We found a strong correlation (Figure 1C, *r* = −0.73, *p* = 0.04), demonstrating that higher relative visual cortex activity indexes more vivid imagery (a lower VVIQ score). This result suggests one can measure visual cortex activity to probe the vividness of a subject’s imagery, thus obtaining a more objective measure of a previously subjective rating” [26] (p. 476).

*STUDY 2: Vividness of visual imagery depends on the neural overlap with perception in visual areas* [27]. I quote from the authors’ abstract, as follows: “Research into the neural correlates of individual differences in imagery vividness point to an important role of the early visual cortex. However, there is also great fluctuation of vividness within individuals, such that only looking at differences between people necessarily obscures the picture. In this study, we show that variation in moment-to-moment experienced vividness of visual imagery, within human subjects, depends on the activity of a large network of brain areas, including frontal, parietal, and visual areas. Furthermore, using a novel multivariate analysis technique, we show that the neural overlap between imagery and perception in the entire visual system correlates with experienced imagery vividness. This shows that the neural basis of imagery vividness is much more complicated than studies of individual differences seemed to suggest” [27] (p. 1327).

“Significance statement: Visual imagery is the ability to visualize objects that are not in our direct line of sight—something that is important for memory, spatial reasoning, and many other tasks. It is known that the better people are at visual imagery, the better they can perform these tasks. However, the neural correlates of moment-to-moment variation in visual imagery remain unclear. In this study, we show that the more the neural response during imagery is similar to the neural response during perception, the more vivid or perception-like the imagery experience is” [27] (p. 1327).

“Results: To directly compare activity between perception and imagery, we contrasted the two conditions (see Figure 3). Even though both conditions activated the visual cortex with respect to baseline, we observed stronger activity during perception than imagery throughout the whole ventral visual stream. In contrast, imagery led to stronger activity in more anterior areas, including insula, left dorsal lateral prefrontal cortex, and medial frontal cortex… We modeled the imagery response for each vividness level separately.” In Figure 4 the investigators plotted the difference between the main effect of perception and the main effect of imagery in the early visual cortex, for each vividness level.

## 5. Action

“I always like to picture the game the night before: I’ll ask the kitman what kit we’re wearing, so I can visualize it. It’s something I’ve always done, from when I was a young boy. It helps to train your mind to situations that might happen the following day. I think about it as I’m lying in bed. What will I do if the ball gets crossed in the box this way? What movement will I have to make to get on the end of it? Just different things that might make you one percent sharper,” said Wayne Rooney [28].

The foundation for one’s mental model of the world is coded in the central nervous system, which, among its many functions, represents the perceived and imagined worlds in various modalities of sensory-affective imagery. Visual imagery is understood to be a quasi-sensory experience which shares at least some of its generating processes with perception [29,30]. It is accepted that the perceptual system is inextricably linked to the action system, such that perceiving something often leads to some corresponding activity, either covert or overt [31]. Perceiving and imaging are not merely processes of identification brought about by looking and listening, but active performances in which specific intentions, purposes, and actions need to be fulfilled [32] (p. 6). In part, this is the distinction between “seeing what” and “seeing as”. What something is seen as has implications for action. This fact is problematic for any theories in which perception and action are not functionally interlinked. If I am feeling thirsty in the desert and I see a reflection as a lake, I will move as rapidly as possible toward it feeling hopeful and encouraged. If I see the reflection as a mirage, I will try to ignore it and continue slowly on my way feeling discouraged. Note that the example includes an affective-motivational component, which is a characteristic of all types of activity and not just a selected example. We arrive at the activity cycle theory (ACT) [32,33] (Figure 5). The arrows in this and all the other figures in this article represent cause-and-effect connections, not correlational associations.

The activity cycle theory of conscious imagery claims that a primary function of consciousness is the mental rehearsal of adaptive, goal-directed action through the experimental manipulation of perceptual-motor imagery [33]. As predicted by this theory, meta-analyses [5,25] showed that the vividness of conscious mental imagery is strongly associated with precisely those performances most likely to benefit from the use of perceptual-motor imagery and mental practice. ACT helps to explain the existence and function of conscious experience.

The image is a cycle of activity triggered by any of four processes: object, affect, schema, and activity. As the evidence shows, the more vivid the image is, the more closely neural activation resembles that activated in real physical activity with actual objects. Covert actions are similar in neural terms to the state of execution of that action overtly. Mental imagery can be as simple as imagined looking, listening, or touching, or as complex as preparing a gourmet dinner or designing a scientific theory. The triggering schemata can be activated top-down or bottom-up. A cycle of mental activity always must include a goal, the means to reach it, and the consequences on the organism and objective world.

Evidence for the affective and somatic components of mental imagery is strong and was discussed for at least a century. In 1907, Wundt [34] wrote the following:

“When any physical process rises above the threshold of consciousness, it is the affective elements which, as soon as they are strong enough, first become noticeable. They begin to force themselves energetically into the fixation point of consciousness before anything is perceived of the ideational elements… They are sometimes states of pleasurable or unpleasurable character, sometimes they are predominantly states of strained expectation… Often there is vividly present… the special affective tone of the forgotten idea, although the idea itself still remains in the background of consciousness… In a similar manner… the clear apperception of ideas in acts of cognition and recognition is always preceded by feelings” [34] (pp. 243–244).

Silvan Tomkins taught us that the primary motivational system is the affective system, and that biological drives have impact only when amplified by the affective system [35]. A similar view was reached by Robert Zajonc [36]. When subjects imagine happy, sad, and angry situations, different patterns of facial muscle activity are produced that can be measured by electromyography [37]. People see objects with feeling. Affective representations of visual sensations are included in the brain’s predictions of what sensations stand for and how to act on them in the future [38]. Similar affective responses occur when people mentally image not only faces, but more complex objects, scenes, and activities; however, the physiological responses are generally less intense in mental images [13].

## 6. Volition

Volition or will is the process by which an individual needs, wants, and commits to a course of action or goal. Volition is purposive striving, a primary psychological function for stability, security, and survival. Unsurprisingly, goal-directedness is at the root of many psychological theories and a significant feature of theories of consciousness [39,40]. In *The Theatre of Consciousness*, Bernard Baars [39] stated the following:

The only conscious components of action are as follows:*a.* the “idea” or goal (really just an image or idea of the outcome of the action);*b.* perhaps some competing goal;*c.* the “fiat” (the “go signal”, which might simply be release of the inhibitory resistance to the goal);*d.* sensory feedback from the action.

Most actions are automatically generated. Only when there is novelty or special care required are actions conscious and under voluntary control. In his ideomotor theory of voluntary control, William James proposed that ideas and images trigger automaticity in brain centers that carry out voluntary actions. Conscious events are only needed to specify new goals and actions. When people focus on their movements, they may actually interfere with the automatic control processes that normally regulate movements, whereas focusing on the effect of movement allows the motor system to self-organize more naturally [41].

Affect, feelings, and drives are intimately related to the sense of striving to satisfy desires and goal attainment. Volition is the driving force of mental image generation and the contents of consciousness [42]. William James stated that “the pursuance of future ends (goals) and the choice of means for their attainment [schema] are the mark and criterion of the presence of mentality (consciousness) in a phenomenon” [42] (p. 8). Goals are set at a meta level governed by drives, values, and beliefs to allow planning of actions, inhibition of actions, and reflection (“wait and see”) as the situation requires. A similar theory was independently developed by Marc Jeannerod (1999) [43].

The evidence suggests that consciousness evolved earlier in human evolution than language [44]. How much earlier is a matter for debate. Feinberg and Mallatt [45] argued that consciousness evolved 520 to 560 million years ago. Developmentally, also, consciousness is accessible earlier than language [44]. For these reasons, language and speech are subservient to imagery. Among its many uses, vivid imagery plays a key role in planning goal-directed behavior. The cognitive system has a meta level to control and monitor the object level. This duality of levels is advantageous because it enables moment-by-moment adjustments to goal-seeking behavior at the object level. Mental imagery instantiates the conscious representation of the self—the “I” of “I am”—and the environment, and interactions between self and others. Conscious imagery allows the conduct of mental simulations of action sequences at the object level without energy expenditure or risk. The object level interfaces with the social level in the public domain of shared activities and object levels. The possible outcomes of alternative future actions may be appraised prior to a course of action. In this way, conscious mental imagery serves as a mental toolbox, producing its internal contents for the user to explore and manipulate in the selection and preparation of future physical and social activity. This idea was anticipated by Francis Bacon more than 400 years ago when he wrote the following:

“For sense sendeth over to imagination before reason have judged; and reason sendeth over to the imagination before the decree can be acted; for imagination ever precedeth voluntary action” [46].

The principal role of mental imagery is to perform “thought experiments” by rehearsing activation of schemata and simulating alternative cycles of action to evaluate potential outcomes in the objective world before making actions physically. Conscious imagery provides the necessary competence to perform thought experiments in a risk-free manner. Thought experiments enable the imager to generate a sequence of interacting processes consisting of goals, schemata, actions, objects, and affects. The envisaged processes are similar to those of the scientist establishing theories and hypotheses and testing them using controlled investigation. The aims, methods, and results of the experiment are at the meta level and are determined by needs, beliefs, and values. The remaining processes are at the object level. Once triggered, implementation of activity cycles gives rise to actual physical activity, perception, and feeling.

ACT describes and explains the mental processes involved in the continuous sequence of adaptations and interactions that occur in making decisions and choices about how to act in the social and physical worlds. Conscious mental imagery serves a basic adaptive function in enabling a person to prepare, rehearse, and perfect his or her actions. Mental imagery provides the necessary means to guide experimentally and transform experience by running activity cycles as mental simulations of the real thing. Such activity rehearsal can only proceed effectively when the rehearsal incorporates vivid imagery. Imagery that is vivid, through virtue of being as clear and as lively as possible, closely approximates actual perceptual-motor activity, and is of benefit to action preparation, simulation, and rehearsal. In cases where conscious imagery is absent or removed by central nervous system (CNS) injury, there is a need for an alternative means of action planning.

It is established that 2–3% of the population has “congenital aphantasia”, meaning that they do not experience visual mental imagery, or lack the ability to control it, but are no less able to control their behavior than people who can visualize. A study of 21 cases of aphantasia found that the majority of participants had some experience of visual imagery from dreams or involuntary “flashes” of imagery, for example, at sleep onset. Thus, their aphantasia involved a deficiency of voluntary imagery rather than a total absence [47]. In one recent study, an aphantasic individual performed significantly worse than controls on the most difficult visual working memory trials [48]. Her performance on a task designed to involve mental imagery did not differ from controls. However, she lacked meta-level cognitive insight into her performance, which is consistent with the current theory. Aphantasic individuals are able to use verbal intentions in goal-directed behavior, schematic spatial imagery, and non-imaged action plans as alternative control systems.

## 7. Consciousness

After millennia of deep thought, a question that continues to baffle philosophers and psychologists is why humans need consciousness. Knowledge concerning vividness helps answer this question. Mental imagery occurs in a wide range of states of consciousness from waking to sleep. It is the tenet of the current theory that sensory-affective mental images are basic building blocks of consciousness in perception, memory, and imagination [32,44].

Building knowledge requires asking questions. Many times, asking a “good” question leads straight to another question, and so on, until finally there is an answer that may be useful to somebody. No psychological topic prompts more questions than consciousness. When I taught a university BSc Psychology honors course on “consciousness” 40 years ago, it was seen as “off-the-wall” and irrelevant. The only thing was that the students loved this subject and my course received higher ratings that the more traditional courses. Apparently, I was ahead of the game. Now, consciousness is mainstream, and more is known, but there is much more still to learn.

*What is consciousness, what is it “made of”, and what is it for?* To answer these questions, it is sensible to consider what we think we mean when we speak about consciousness and to work from there. I list here 30 claims, indicating which are part of the meta level of executive control (*items in italics*).
*(i)* Consciousness is agentic, i.e., it has purpose, desire, and intentionality;*(ii)* It is deeply social in nature;(iii) It is the center for feelings and moods;*(iv)* It operates with an inbuilt motivation to drive the organism toward pleasure and away from pain;*(v)* It is a center for perceptions, interoceptive and exteroceptive;*(vi)* It serves as a “storehouse” of memories including autobiographical memories from which information and images can be retrieved;*(vii)* It is the control center for action, perception, attention, affect regulation, cognition, and information processing, all of which require the making of predictions;*(viii)* It has “layers” and “levels” and is capable of dissociation, splitting, and confusion;*(ix)* It constructs a personal and a public identity for the “self”;*(x)* It is a center for constructing and changing values and beliefs;*(xi)* It can set both altruistic and selfish goals, and anything in between;*(xii)* It can represent information, beliefs, and values in an honest way, or it can simulate, pretend, lie, and be deceitful;*(xiii)* It can be subject to hearing of voices and other hallucinations;*(xiv)* It can be subject to illusions and delusions;*(xv)* It can be accessed by introspection;*(xvi)* It can be described symbolically in speech, writing, and in works of art, but it can also be ineffable;*(xvii)* It varies in state of arousal from waking to sleep;*(xviii)* It references values, beliefs, rules, and customs, and has pragmatic methods for following them;*(xix)* It strives for the satisfaction of needs including equilibrium;*(xx)* It can pay close attention to detail or its concentration can wander;*(xxi)* It fantasizes and “daydreams”;*(xxii)* It plans new goals for the future;*(xxiii)* It thinks and makes decisions;*(xxiv)* It imagines and weighs consequences pro and con before acting;*(xxv)* It receives feedback on the outcomes of action;*(xxvi)* It “delegates” well-practiced routines, tasks, and habits to a lower level of automatic processing;*(xxvii)* Automatic functioning, such as the autonomic system, is also below the threshold of consciousness as long as it is performed as expected, but it becomes conscious if it fails to perform normally;*(xxviii)* It dreams;*(xxix)* It maintains type II homeostatic responses of the whole organism;*(xxx)* It remains imperfect.

Based on the above list, a principle of consciousness (PC) can be stated as follows:***Consciousness is the central executive process of the brain that builds images of the world, makes predictions about future events, and selects which voluntary actions to execute.***

The major inputs to consciousness are exteroceptive, sensory stimuli—sight, sound, taste, smell, touch, temperature, vibration, and pain—and also interoceptive stimuli, which form a cortical image of homeostatic afferent activity from the body’s tissues. This system provides experiences and visceral feelings such as pain, temperature, itch, sensual touch, muscular and visceral sensations, vasomotor activity, hunger, thirst, and “air hunger”. Interoceptive activity is represented in the right anterior insula, providing subjective imagery of the material self as a feeling (sentient) entity, that is, emotional awareness [49]. As the PC states, one of the outputs of consciousness is something that human beings could not possibly do without: predictions. Predictive simulations, otherwise known as rehearsals, involve “what-if” or “if-then” relationships: “If I do X, will Y or Z happen”.

Anything that happens between stimulus input and response output is based on if-then operations and simulations geared toward stability and safe prediction. Private fantasies and daydreams take up at least a half of our waking time. It is known that there is a huge quantity of pre-conscious automatic processing of sensory information and behavior that does not require the effortful attention of consciousness. The controlled processing of consciousness is serial, attention-demanding, methodical, and slow, e.g., preparing a meal using a cookery book or reading a manual on how to operate a digital versatile disc (DVD) player. Automatic processing, on the other hand, is efficient and economical, and, quite often, quick, e.g., reading, writing, walking, riding a bicycle, or driving a car.

Brain science supports the idea that the forebrain of the cerebral cortex is the site of the central control system of consciousness. The forebrain itself is involved in regulation of both autonomic and non-autonomic human responses in stress and affect. The forebrain is also the seat of both type I and type II homeostasis.

The significant part of the content of consciousness is mental imagery, the quasi-perceptual mental imagery that gets us from one point on our mental model of the world to the next. Before turning to explore the nature and function of mental imagery, it is essential to say more about how imagery fits into the system as a whole.

## 8. Psychological Homeostasis

The 19th century French physiologist, Claude Bernard, “the father of modern physiology and experimental medicine”, is best known for his work on the pancreas and vasomotor system, and for discovering glycogen. Yet, his description of the “milieu intérieur” in living organisms is equally significant.
***“The stability of the internal environment is the condition for the free and independent life.”***

Bernard’s [50] “milieu intérieur” concept was ignored for many decades; nobody really knew what to do with it. Then, in the early 20th Century, J.S. Haldane, C.S. Sherrington, J. Barcroft, and a few others started to work with it. In 1926, the Harvard physiologist Walter Cannon translated the French term into Greek and coined the term homeostasis [51]. Cannon thought that the automatic function of homeostasis freed the brain for more intellectual functions such as intelligence, imagination, insight, and manual skill.

Homeostasis is necessary for every living system and could be the defining characteristic of life itself. At every level of existence, from the cell to the organism, from the individual to the population, and from the local ecosystem to the entire planet, homeostasis is a drive toward stability, security, and adaptation to change. In an infinite variety of forms, omnipresent in living beings, is an inbuilt function with the sole purpose of striving for equilibrium, not only in the “milieu intérieur” but also in the “milieu extérieur”. On the other side of Bernard’s scientific coin, I postulate the following basic principle:
***“The stability of the external environment is the condition for the free and independent life.”***

By changing a single word “internal” to “external”, one creates a whole new theoretical perspective for consciousness, volition, cognition, affect, and behavior, a General Theory of Behavior based on the construct of homeostasis. Striving for balance and equilibrium is the primary guiding force in all that we plan, think, feel, and do. I call this homeostasis (type II) the “reset equilibrium function” (REF) [1].

Every organism automatically regulates essential physiological functions by homeostasis, and internal drives are maintained in equilibrium using corrective behavior in the form of eating, drinking, defecating, sleeping, and so on. This form of homeostasis was established scientifically since the time of Bernard and was implicit as a concept within classical theories of Hippocrates. Far more than this, without any special reflection in most instances, all conscious beings are constantly and quite routinely reconciling the discrepancies among their thoughts, behaviors, and feelings, and in the differences with those with whom they have social relationships. Conscious organisms strive to achieve their goals while maximizing cohesion and cooperation with both kith and kin and, at the same time, strive to take away or minimize the suffering and pain of others. The goal is to minimize all forms of tooth-and-claw competition to live in a culture where the thriving of all is in the self-interest of every individual. This idea was described by Antonio Damasio [9] as “cultural instruments first developed in relation to the homeostatic needs of individuals and of groups as small as nuclear families and tribes. The extension to wider human circles was not and could not have been contemplated. Within wider human circles, cultural groups, countries, and even geopolitical blocs often operate as individual organisms, not as parts of one larger organism, subject to a single homeostatic control. Each uses the respective homeostatic controls to defend the interests of its organism” [9] (p. 32).

Aware of it or not, the REF is omnipresent; wherever one goes and whatever one is doing, the REF is jogging along with us every step of the way. The REF is not something one focuses attention on, but it is nevertheless the process by which our behavioral systems are perpetually striving to maintain balance, safety, and stability in our physical and social surroundings. Competing drives, conflicts, and inconsistencies can all pull the flow of events “off balance”, triggering an innate striving to restore equilibrium.

For the majority of time, the majority of people strive to calm and quieten disturbances of equilibrium rather than to acerbate them. It is not a battle that is always won; there is always the possibility of instability, calamity, or catastrophe even. If one cannot win every battle, one can at least strive to win the war.

Courtesy of homeostasis, body and mind are continuously regulating and controlling in multiple domains and levels simultaneously, with constant resets and automatic adjustments to both voluntary and involuntary behavior. Type I homeostasis is the inwardly striving physiological homeostasis H[Φ] and type II homeostasis is the outwardly striving psychological homeostasis H[Ψ]. From birth to death, these two forms of homeostasis provide optimum levels of controllable equilibrium. The reset equilibrium function (REF) integrates the principle of homeostasis with our understanding of psychological processes and behavior. Systems theory with cyclical negative feedback loops is a central feature. Feedback loops in cybernetics and control theory mirror homeostasis within biology and neuroscience. Psychologists employ control theory as a conceptual tool for large areas of psychology [52]. Notably, one objective of control theory was always to provide a “unified theory of human behavior” [53]. A unified theory integrates knowledge about consciousness with knowledge about behavior.

## 9. The General Theory of Behavior

This General Theory draws upon systems of homeostasis consisting of interconnected processes in continuous feedback loops that are updated with each reset of the REF. The REF extends the reach of homeostasis to a general control function which automatically restores psychological processes to equilibrium and stability. The REF is triggered when any of the processes within a system strays outside of its set point or range. The REF is innate, but it can only exist in conscious organisms which all have two kinds of homeostasis (types I and II). Non-conscious organisms are availed with only one type of homeostasis (Type I). Type II homeostasis exists in a system with any number of processes, each with its own set range, making a series of resets.

Any set of processes, such as the four shown in Figure 6, is a tiny sub-set of thousands of interconnected processes responsible for coding and communicating inside the body and the brain. Any process can be connected to hundreds or thousands of other processes, any one of which can push any particular process out of its “comfort zone”, thereby requiring it to reset. As any one process resets, a “domino effect” among many other interconnected processes requires these to reset also. Thus, a reset is often a complete reset of a large part of the entire system, not simply the resetting of a single process. Psychological and physiological processes operate in tandem to maximize equilibrium for each particular set of functions.

The General Theory explains the relevance of the REF to numerous psychological functions including those where reset is a condition for change, e.g., affect, chronic stress, excessive behaviors such as smoking, drinking, gambling, and overeating, pain, sleep loss, and low subjective well-being. In all of these situations, the subject’s conscious acknowledgement that there is behavior in need of change is of primary importance. This acknowledgement is a necessary precondition for purposeful striving toward making that change.

“The purpose of a brain is not to think, but to act” [54]. The central executive system of consciousness enables organisms to mentally map the environment, predict what might happen next, and to act. One of the major processes for modeling, predicting, and acting is mental imagery. Mental imagery is ideally suited to these purposes by providing preparatory images, which can exist in any sensory modality; however, for the majority of people, this is predominantly visual. On the other hand, imagining the smell and taste of a delicious meal, “hearing” the sound of some enchanting music, and imagining scenes and feelings of relaxation from a recent holiday, or, indeed, “tasting” a delicious glass of wine are all equally possible [55]. Anomalous and paranormal experiences are reported at significantly higher rates among people with vivid VVIQ scores, especially auras, remote healing, and apparitions, but only among high vivid scorers in the open-eyes condition [56].

According to Frederic Bartlett (1932), schemata are much more than elementary reactions ready for use; “they are also arrangements of material, sensory at a low level, affective at a higher level, imaginal at a higher level yet, even ideational and conceptual” [31]. The action system is inextricably linked to the perceptual system, such that perceiving something generally leads to activity in either covert or overt form triggered by schemata. Imagined simulation consists of covert performances in which specific intentions, purposes, and actions are fulfilled [32]. Mentally simulating an experience serves as a substitute for the corresponding experience [57]. Based on this formulation, a mental imagery principle can be stated as follows:
***Mental imagery principle: A mental image is a quasi-perceptual experience that includes action schemata, affect, and a goal.***

A system based on this principle is shown in Figure 7.

Involuntary images, persistent, unpleasant memories, and repetitive habits are symptomatic of disorders, e.g., patients with posttraumatic stress disorder, anxiety disorders, depression, eating disorders, and psychosis frequently report repeated visual intrusions concerning real or imaginary events that can be extremely vivid, detailed, and with highly distressing content [58]. Hallucinations are of particular interest because they are reported by much larger numbers of people than those who have diagnoses of psychosis, and there is a significant overlapping in phenomenology with subjective paranormal experience such as precognition and out-of-the-body experience. One definition of hallucination states it to be “any percept-like experience which (a) occurs in the absence of an appropriate stimulus, (b) has the full force of impact of the corresponding actual (real) perception, and (c) is not amenable to the direct or voluntary control of the experiencer [59] (p. 23). In the first two parts, hallucination is a form of mental imagery. The third part provides the distinguishing feature because, to be beneficial, mental images need to be voluntary and controllable by the experiencer. The full spectrum of conscious experience, including dissociative states, psychotic episodes, hallucination, pseudo-hallucinations, out-of-the-body experiences, delusions, ipseity, subjective paranormal experience, and the varying ability to exert voluntary control, is the subject of another paper [60].

As noted, the General Theory proposes a cyclical system of objects, schemata, affective experience, and actions. The control system has both an executive level and a schema level. The executive level, which is what is normally referred to as “consciousness”, controls and monitors the schema level. This duality of levels enables moment-by-moment adjustments to goal-seeking behavior at the schema level. Goals are set at the executive level of consciousness. Goal-setting is guided by values and beliefs which inform actions, inhibit actions, or reflect on what action to take, as the situation requires.

In competent performers, speech, decisions, routines, and many complex behaviors normally do not require conscious control to operate [61]. Afferents from the muscles and the activity of the cerebellum, where movement is organized, operate entirely preconsciously and produce no conscious images [62,63].

Conscious imagery is useful in the planning and organization of behavior through enabling the simulation of action sequences at the object level without energy expenditure or risk. The object level interfaces with the social level in the public domain of shared activities and object levels. The possible outcomes of alternative future actions can be appraised prior to a course of action. In this way, conscious mental imagery serves as a mental toolbox, producing its internal contents for the user to explore and manipulate in the selection and preparation of future physical and social activity.

## 10. The Clock System

An internal clock controls physiological and behavioral processes of daily living in synchrony with regular changes in the environment. Over hundreds of millions of years in an environment that changes dramatically over every 24-h cycle, evolution produced universal rhythms throughout the plant and animal kingdoms such that each organism’s biochemistry, physiology, and behavior are organized in diurnal cycles [64]. Many circadian rhythms are persistent even in the absence of the normal diurnal cues of night and day or temperature changes, e.g., while living in caves. Such demonstrations are interpreted as reflecting the operation of an internal biological clock or clocks. The circadian clock system serves as a biological “alert” that lets us know when significant events are due to happen.

The light-dark (LD) cycle is the most reliable of the external signals enabling entrainment and is referred to as a “zeitgeber” (i.e., time-giver). LD information is perceived by mammals with retinal photoreceptors and conveyed directly to the suprachiasmatic nucleus (SCN) of the hypothalamus, where it entrains oscillators in what is regarded as the master clock of the organism [65]. Other cyclic inputs, such as temperature, noise, social cues, or fixed mealtimes, also can act as entraining and predictive agents, although usually to a less reliable extent than LD.

An entrainable circadian clock is present in the SCN during fetal development, and the maternal circadian system coordinates the phase of the fetal clock to environmental lighting conditions. Even before birth, the organism is entrained to the LD cycle [66]. Having a clock system is advantageous for predicting and preparing for important events. When food is available only for a limited time each day, it was observed that rats increase their locomotor activity two to four hours before the onset of food availability [67]. Similar anticipatory behavior occurs in other mammals and in birds, accompanied by increases in body temperature, adrenal secretion of corticosterone, gastrointestinal motility, and activity of digestive enzymes.

It was proposed that a common design principle applies to the clock in all organisms, from bacteria to humans, and that the circadian clock existed for at least 2.5 billion years. [64]. The predictive mechanism in which physiology and behavior are “tuned” to the timing of external events allows a competitive advantage. When disrupted by genetic or environmental means, cardio-metabolic diseases and cancer can be triggered, and realigning out-of-sync circadian rhythms can be beneficial in the treatment of endocrine-related disorders [67].

Synchronicity at a neural level with 35–75-Hz oscillations in the cerebral cortex, hypothesized to be “the basis of” consciousness, form the binding that may be achieved by the synchronized oscillations of neuronal groups [68]. It is suggested that two items of information (e.g., shape and color) are bound together if the relevant neural groups oscillate with the same frequency and phase. Consciousness, I would humbly suggest, is an emergent property of evolution.

## 11. The Approach Avoidance Inhibition (AAI) System

“Every person on the planet (barring illness) can tell good from bad, positive from negative, pleasure from displeasure” [69]. Not only can one tell it, one can feel it also. From the pre-Socratic philosophers until the present day, the role of pleasure and pain as motivators of human behavior is universally accepted. Psychological hedonism, the idea that all action is determined by the degree of pleasure or displeasure that imagining the action provokes, dates back to Epicurus (341–270 BC)), who is alleged to have said “we begin every act of choice and avoidance from pleasure…” [70]. The idea that organisms strive for pleasure and the avoidance of pain was accepted for eons. Michel Cabanac suggests that the pleasure or displeasure of a sensation is directly related to the biological usefulness of the stimulus [71]. The seeking of pleasure and the avoidance of displeasure have useful homeostatic consequences. That is, they depend on the internal state of the stimulated subject at the particular moment of the stimulation. Pleasure indicates a useful stimulus and motivates approach, while pain indicates a useful stimulus and motivates avoidance.

Emerging evidence indicates similarities in the anatomical substrates of painful and pleasant sensations in the opioid and dopamine systems [72]. The experiences of positive and negative affect are based on neural circuits that evolved to ensure survival. These circuits are activated by external stimuli that are appetitive and life-sustaining or by stimuli that threaten survival. Activation of the pain and pleasure circuits alert the sensory systems to pay attention and prompt motor action [72]. The approach-avoidance concept was pivotal in theories of behavior [73]. The approach-avoidance system includes behavioral inhibition which takes over when there is approach-avoidance conflict. The approach-avoidance-inhibition (AAI) system sets the bar for fight-fright-freeze decisions that are pervasive throughout the animal kingdom. Action schemata are necessary precursors to action in a four-pronged system for regulating approach-avoidance-inhibition (AAIS). Operating together with action schemata, the REF, clock, and AAIS regulate voluntary action (Figure 8).

In the following section, the different modular systems described for imagery, timing, action, and inhibition are integrated into a single system for behavior control.

## 12. Behavior Control System

As the executive controller of brain and behavior, consciousness occupies pole position, providing a significant evolutionary advantage to the organism. Figure 9 shows the behavior control system for the planning and execution of behavior. The system for the regulation of emotion [74] is confined here to a single module for affect. It is accepted that consciousness is the coordinated activity of the brain system as a whole, a direct emergent property of cerebral activity [75]. Consciousness exerts supervening control over the entire flow pattern of cerebral excitation. The system shows large modular functional entities with a huge range of special qualities and properties. The modules interact causally with one another as homeostatic entities to produce the entire gamut of images, feelings, thoughts, and actions, all working toward stability and equilibrium. Mental imagery is a benefit to the planning and prediction roles of consciousness, to the function of psychological homeostasis. If vividness is entirely lacking, then language provides an alternative route (e.g., “first do X, then do Y”), such that, by whatever means, psychological homeostasis can ensure that equilibrium with the environment is maximized at all times.

Critics claim that consciousness is an epiphenomenon and that its contents are formed “backstage” by non-conscious systems. Oakley and Halligan [76] stated that “psychological processing and psychological products are not under the control of consciousness… All ‘contents of consciousness’ are generated by and within non-conscious brain systems in the form of a continuous self-referential personal narrative that is not directed or influenced in any way by the ‘experience of consciousness’”. The behavior control system contains a meta level, a schema level, and an automatized level. It is the executive meta level of consciousness that sets goals and directs the lower levels to prepare and execute actions, and make necessary adjustments. Conscious mental images are also put into service at the schema level, e.g., in simulation, skills, design, drawing, writing, geometry, and other creative performance. The majority of behavior, which can be classed as routine, falls within the automatized level and does not require consciousness. The behavior control system enables decisions about outcomes of alternative future actions to be weighed and appraised prior to committing to any course of action. In addition to language skills, conscious imagery serves as a mental “toolbox”, providing its internal contents for the user to select and prepare future physical and social activity. It is the meta level of consciousness, having the quality of ipseity, an awareness of a unique personal, meaningful point of view, driven by values, beliefs, feelings, desires, and wants, by the universal striving of psychological homeostasis, that sets new projects, monitors progress, and directs the schema level to prepare and execute actions.

According to a personal narrative account [76], the experience of consciousness is telling stories to ourselves about a fictive internal state of “consciousness”. I have strong doubts about this account because it leaves so much unexplained. How does this account explain the experience of pain [77]; why must researchers be given ethical guidelines for investigations of experimental pain in conscious animals? [78]; why does the science of psychology require the concept of dreams [79] and hallucinations? [80]; how are thought experiments employed to create scientific theories? [81]; why are there objective performance gains from mental rehearsal by elite athletes, footballers, and others? [82]; why should differences in visual perspectives, “external” (third-person) and “internal” (first-person), be differentially interfered with by a concurrent action dual task? [83]; why, in reading poems, “vividness of imagery was the strongest contributor to aesthetic pleasure, followed by valence and arousal”? [14]; why is vividness of mental images associated with a stronger sense of presence felt in experiencing virtual reality scenarios? [84]; and why do visualization, first-person perspective, and narratives representing real experiences improve memory and comprehension? [85]. Dozens more examples of how consciousness acts as a meta-level controller of behavior are available. There would be far too many mysteries left to explain if consciousness was simply a figment of a personal narrative.

Another argument against the view that consciousness controls action and thought is the empirical evidence that the outcome of a decision can be encoded in brain activity of prefrontal and parietal cortex seconds before it enters awareness. It is presumed that this delay reflects the operation of a network of high-level control areas that begin to prepare an upcoming decision long before it enters awareness [86]. We need to look no further for an explanation than to Libet himself: “Although the volitional process may be initiated by unconscious cerebral activities, conscious control of the actual motor performance of voluntary acts definitely remains possible. The findings should, therefore, be taken not as being antagonistic to free will, but rather as affecting the view of how free will might operate. Processes associated with individual responsibility and free will would ‘operate’ not to initiate a voluntary act but to select and control volitional outcomes.” [86] (p. 538). My General Theory specifies a level of operations for automatized actions such as tying one’s shoe laces, riding a bicycle, or visiting the bathroom, a category within which Libet’s task clearly falls (Figure 9). The last thing consciousness needs is to be aware of every single little detail in one’s thoughts, actions, and feelings. What it definitely must have at its disposal is the personal self, with purposes, desires, and intentions. There were many criticisms of the original studies by Libet [86]. As Bridgeman, the first of multiple commentators on the Libet study, pointed out, “a careful analysis of the experimental conditions reveals that the subjects’ wills were not as free as the Libet article implies, for the small, sharp movements that they were instructed to make were not freely willed but were requested by the experimenter. The will of a subject was no more free in this design than in reaction-time experiments; the only difference between this experiment and the latter paradigms is that the instruction and the movement are decoupled in time. While performing the task, the subjects do nothing more than obey the instructions” [87] (p. 540).

The authors of another Libet-style readiness potential study [88] suggested “free will is an illusion” because they found evidence of high-level control areas beginning to determine an upcoming decision 10 s before entering awareness. The task consisted of pushing a button using an index finger under a prescription by the investigators to choose “freely”. The task was basic, routine, automatized, and under the control of another person’s demands. A free human operator would be “tied up in knots” if all the minute details of such an elementary task needed conscious, volitional control. Nature was kind when it took such automatized actions out of conscious control. Mechanisms for automatic control mechanisms are on a different level to volitional control. For example, when picking up an object, one tends to grasp at contact points that allow a stable grip. Appropriate grasp points can be re-selected during an ongoing movement in response to unexpected perturbations of the target object, suggesting that automatic control mechanisms guide the fingers to appropriate grasp points distinctly from those involved with volitional control [89]. Volitional control with meta-level consciousness is only required when a new purpose for a new project is formulated. The key point is that new project has a personal meaning and coherence within the life-space of a unique individual. Consciousness is required only when skilled actions are to be executed, actions that might interfere with actions already ongoing, including those of others (Figure 9). Feeling, meaning, intention, and a sense of timing and flow are important influences on the quality and energy of actions. Feeling is integrated with the meaning and purpose of an action through the meta system of consciousness.

Some critics claim that totally unconscious living beings could conceivably exhibit all of the adaptive behaviors of conscious beings, the so-called “zombie” option. They suggest that it is entirely conceivable and coherent to state that all structural and functional characteristics of living beings could be explained without any accompanying conscious experience. It is metaphysically possible that such unconscious zombies could be “in the dark” and be equally effective. This is the “zombie” hypothesis of conscious inessentialism in the philosophy of the mind [90]. The zombie is a fascinating hypothetical concept and nobody has ever yet claimed to know, be or to have built a zombie. If so, it would be suitably undone. The acid test would be to give it/her/him an fMRI while answering the items of the VVIQ, or to examine the zombie’s electroencephalogram (EEG) and eye movements during sleep. Any mutterings about images or dreams could be quickly disposed of when the brain scans failed to show the requisite patterns of activation. The zombie is a freak, never to be found in Nature.

Why did consciousness evolve? I suggest the answer lies in the significant evolutionary advantages of an emergent, purposeful, and integrative process of psychological homeostasis to direct holistic control [9,75]. The case for consciousness as an emergent property was made by Roger Sperry in 1969:

“The long-standing assumption in the neurosciences that the subjective phenomena of conscious experiences do not exert any causal influence on the sequence of events in the physical brain process is directly challenged in this current view of the nature of mind and the mind-brain relationship. A theory of mind is suggested in which consciousness, interpreted to be a direct emergent property of cerebral activity, is conceived to be an integral component of the brain process that functions as an essential constituent of the action and exerts a directive holistic form of control over the flow pattern of cerebral excitation”[75] (p. 532)

Consciousness has an explanatory role in the behavior of organisms through its equilibrium-creating process of meta-level control. An experience that is re-represented in the mind has conscious meta cognition, or self-reflection. The intimate connection between the self, intentionality, and purposive goal-seeking behavior has the pre-eminent advantage of ipseity. Only a conscious being can have the invaluable quality of selfhood, the implicit first-person quality of consciousness that all experience articulates from a first-person perspective as “*my*” experience [91]. The zombie can know nothing of this.

## 13. Homeostasis as a Unifying Concept

Homeostasis is a unifying concept across the disciplines of consciousness studies cognitive neuroscience, psychology, and biology. Yet, it is a much neglected concept. Recent studies of the brain’s anatomical connectivity or “connectome” show that the CNS is at once more complex and more simple than previously assumed. Regions of interest produce coherent fluctuations in neural activity and distributed patterns of activation or networks. Neurobiological networks occur at different organizational levels from cell-specific regulatory pathways inside neurons to interactions between systems of cortical areas and subcortical nuclei. Architectures which support cognition, affect, and action are normally found at the highest level of analysis [92]. Brian Edlow and colleagues investigated the limbic and forebrain structures that form a “central homeostatic network” (CHN) [93] responsible for autonomic, respiratory, neuroendocrine, emotional, immune, and cognitive adaptations to stress. These structures include the limbic system in shared participation in homeostasis. Recent research focused on homeostatic forebrain nodes which receive sensory information concerning extrinsic threats and intrinsic metabolic derangements from the brainstem, resulting in arousal from sleep, heightened attention, vigilance during waking, and visceral and somatic motor defenses. These findings suggest that homeostasis is mediated by ascending and descending interconnections between brainstem nuclei and forebrain regions, which together regulate autonomic, respiratory, and arousal responses to stress. The role of the limbic system in the regulation of homeostasis is being recognized, and the limbic system was added to the central autonomic network of “flight, fight, or freeze”. These findings suggest that homeostasis of type I H[Φ] and type II H[Ψ] are controlled by a single executive controller in the forebrain.

That the forebrain controls both types of homeostasis supports the contention that homeostasis is a unifying concept across biology and psychology. Studies of aging suggest that homeostatic dysregulation proceeds in parallel in multiple physiological systems [94]. Aging is characterized by marked reductions in functional correlations within higher-order brain systems [95]. Physiological and psychological homeostasis can be modeled in the same way as a system of adjustments through a network of connected processes or states [1].

## 14. Conclusions

Consciousness is an open system having many relations to its mental, physical, and social surroundings. Changes in these surroundings produce internal “disturbances” of the system that require adjustment, adaptation, or correction. As originally described by Bernard [50] and Cannon [51], and more recently by the author [1], such disturbances are normally kept within set limits, because automatic adjustments are brought into action such that the internal and external conditions are held fairly constant. Everything known about the executive role of the forebrain in action planning and decision-making and the recently discovered central homeostatic network [93] suggests that this must indeed be the case.

The General Theory of Behavior holds that the reach of homeostasis extends well beyond physiology into many realms of psychology and into society as a whole. Homeostasis type I, H[Φ], and type II, H[Ψ], serve identical stabilizing functions internally in the body and externally in socio-physical interactions, respectively. With Cannon, I hypothesize that “steady states in society as a whole and steady states in its members are closely linked” [51]. H[Φ] and H[Ψ] exist in a complementary relationship of mutual support.

A much neglected topic in the history of psychology and today in consciousness studies is mental imagery. Yet, the evidence suggests that mental imagery is the basic building block of consciousness [33]. In protecting stability, security, and equilibrium in the socio-physical world, psychological homeostasis is one of the primary functions of consciousness, and could not exist without it. The emergence of homeostasis and consciousness in evolution would not have been possible without mental imagery. It is impossible to say which came first, consciousness or psychological homeostasis. One thing of which we can be certain is that each of us can truthfully say: “I am conscious; therefore, I am”.

## Figures and Tables

**Figure 1 brainsci-09-00107-f001:**
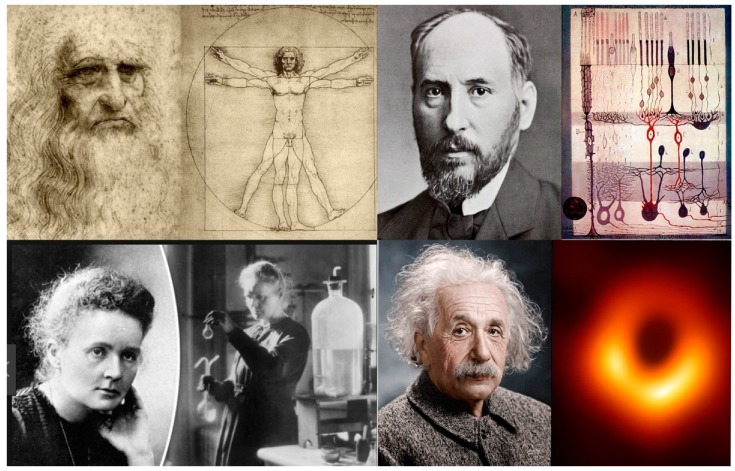
Leonardo da Vinci, Ramón y Cajal, Marie Curie, and Albert Einstein—creative people who used vivid mental imagery to make world-changing discoveries. Einstein’s thought experiments and his statements on the imagination are particularly salient.

**Figure 2 brainsci-09-00107-f002:**
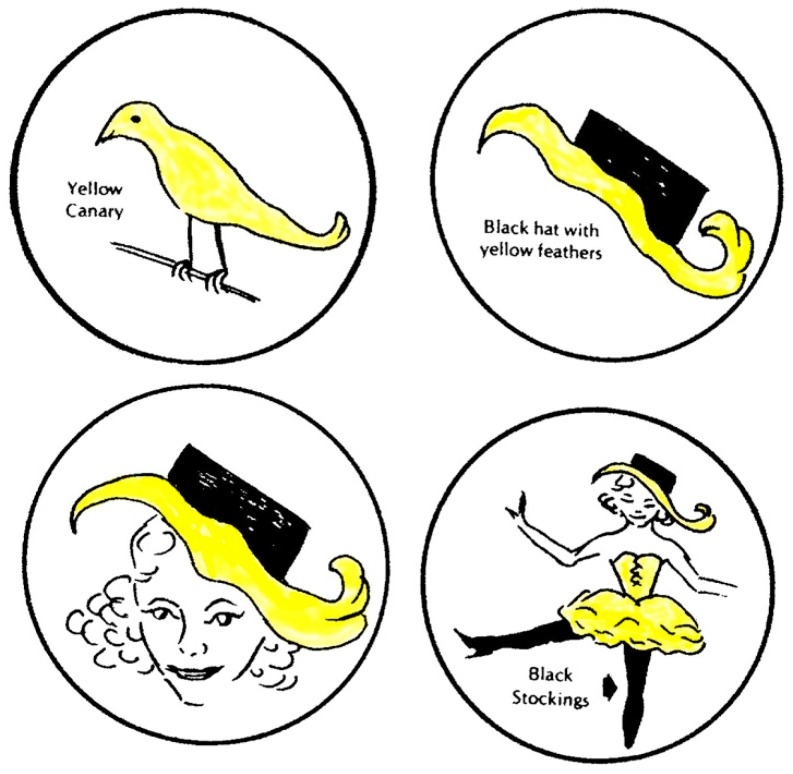
Sequence of projected visual images in response to the suggestion “yellow”.

**Figure 3 brainsci-09-00107-f003:**
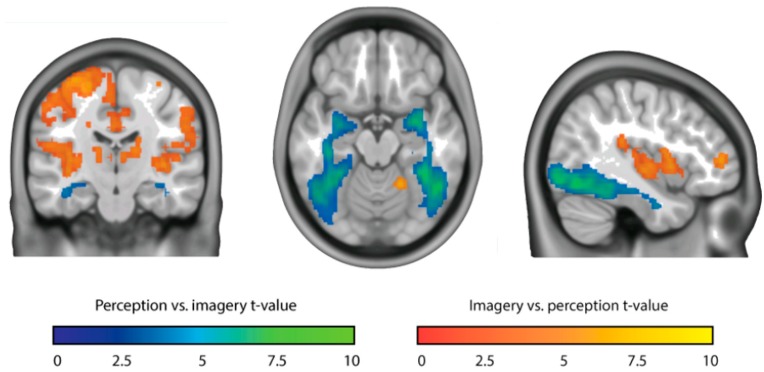
Perception versus imagery. Blue-green colors show *t*-values for perception versus imagery and red-yellow colors show *t*-values for imagery versus perception. Shown *t*-values were significant at the group level. Even though both conditions activated the visual cortex with respect to baseline, stronger activity occurred during perception than imagery throughout the whole ventral visual stream. In contrast, imagery led to stronger activity in more anterior areas, including insula, left dorsal lateral prefrontal cortex, and medial frontal cortex. Reproduced from Reference [27] with permission.

**Figure 4 brainsci-09-00107-f004:**
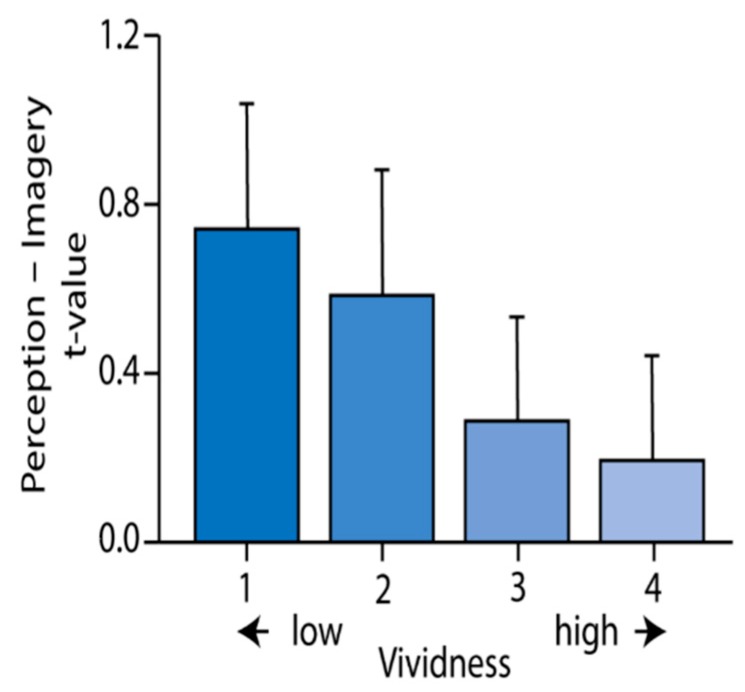
Difference between the effect of perception and the effect of imagery, separately for the four vividness levels. The higher the vividness is, the lower the difference between imagery and perception will be. In each trial, participants were shown two objects successively, followed by a cue indicating which of the two they subsequently should imagine. During imagery, a frame was presented within which subjects were asked to imagine the cued stimulus as vividly as possible. After this, they indicated their experienced vividness on a scale from one to four, where one was low vividness and four was high vividness. The results are shown for a voxel in the early visual cortex that showed the highest overlap between the main effect of perception and the main effect of imagery. More vivid imagery was associated with a smaller difference between perception and imagery [27] (p. 1335). Reproduced from Reference [27] with permission.

**Figure 5 brainsci-09-00107-f005:**
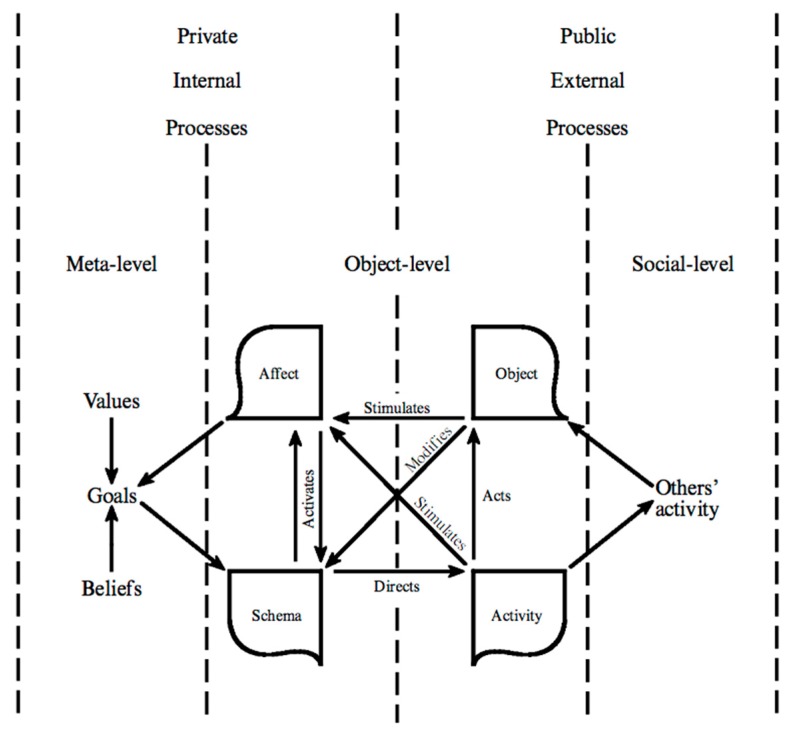
Activity cycle theory (ACT) consists of four systems, three representing a functional module for a psychological domain (affect, schema, activity) and one (object) representing physical objects or internal images and ideas. Private (internal) and public (external) processes are executed across the meta level, object level, and social level of organization. The system enables outcomes of alternative future actions to be appraised prior to committing to any course of action. Conscious imagery serves as a mental “toolbox”, providing its internal contents for the user to explore in the selection and preparation of future physical and social activity. The affect system co-activates goals and schemata enabling the organism to strive toward desired outcomes. Please note that psychological homeostasis was not included in the ACT. However, the ACT formulation was a crucial step in formulating the General Theory of Behavior. Reproduced from Reference [33] with permission.

**Figure 6 brainsci-09-00107-f006:**
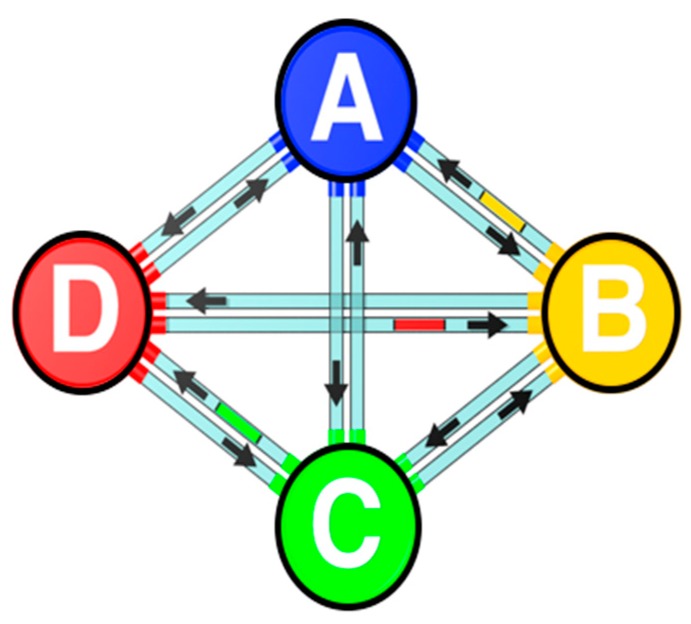
A network of four interconnected processes (A–D) in homeostasis. The reset equilibrium function (REF) returns each process in to its set range. An adjustment in one process may necessitate a compensatory adjustment in one or more of the other interacting processes until equilibrium is reached. Thus, when A stimulates B to lower its activity level, the reduced value in B stimulates C, D, and A.

**Figure 7 brainsci-09-00107-f007:**
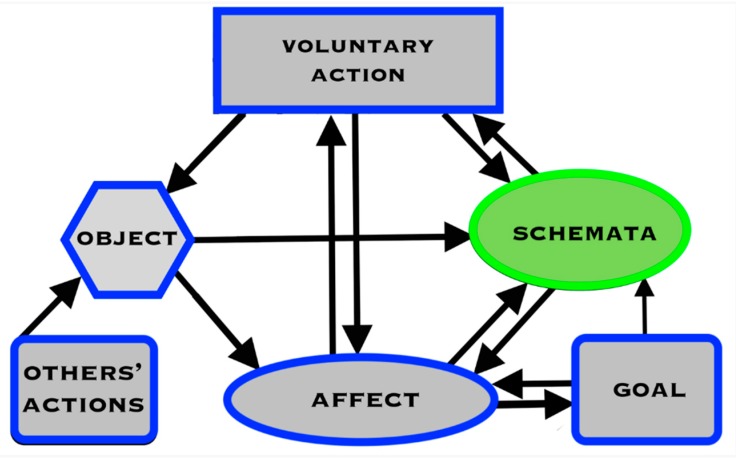
A model of voluntary action (‘VOAGA’ model). Action schemata (As) control voluntary actions (V) in response to salient objects (O) in the immediate environment, in accordance with current goals (G). Affect (Af) influences the goal and the schemata. Action simulation using mental imagery occurs in the same system as that used for overt action.

**Figure 8 brainsci-09-00107-f008:**
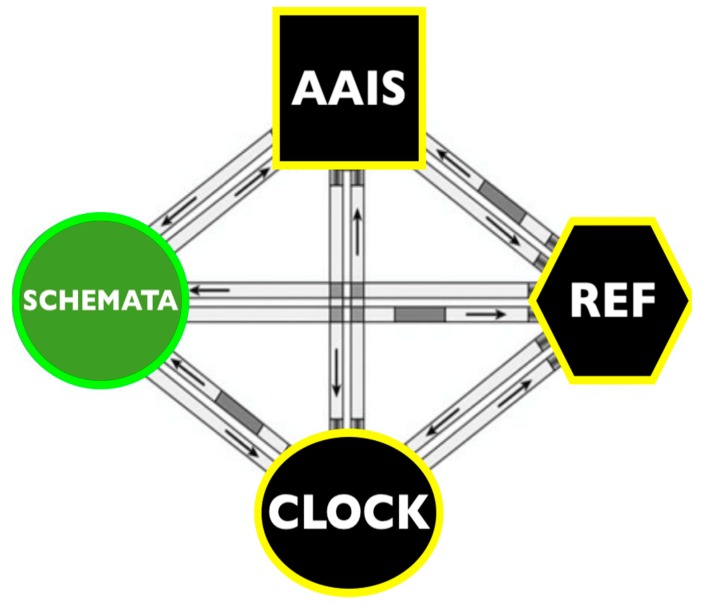
The reset equilibrium function (REF), clock, and approach-avoidance-inhibition system (AAIS) interconnect with action schemata to regulate voluntary action.

**Figure 9 brainsci-09-00107-f009:**
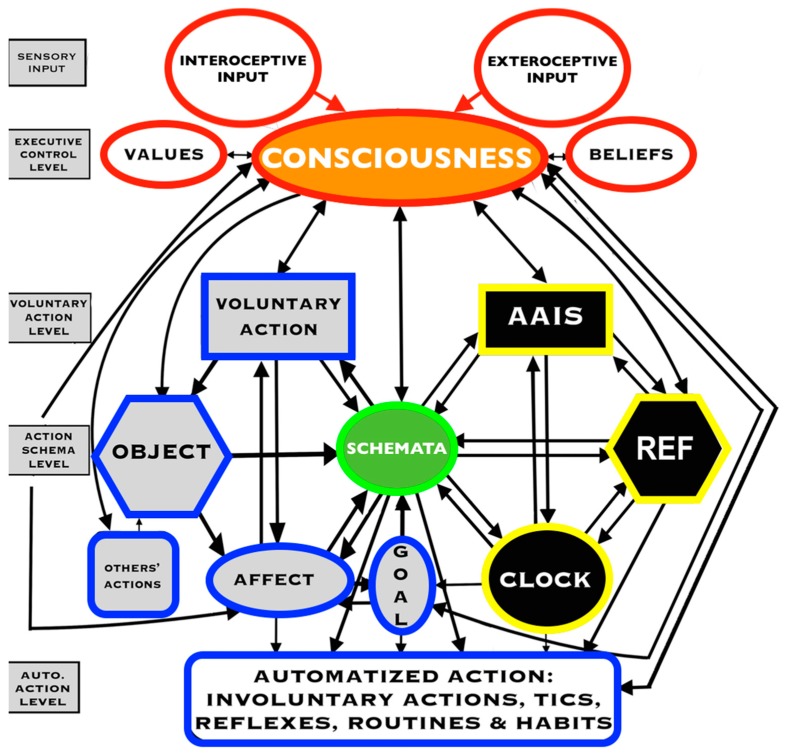
The behavior control system shown with five levels of organization and nine modular systems for the generation of action with consciousness at the executive level. The modules fall into three groups: (i) the ACT group of modules from Figure 7 (gray and blue); (ii) the REF (type II homeostasis), clock (circadian system), and AAIS (approach/avoidance/inhibition system) from Figure 8 (black and yellow) interconnecting with the ACT group via action schemata (green); (iii) the behavior control system/consciousness (red, white, and orange) includes modules for interoceptive and exteroceptive sensory input, beliefs, and values. Five levels of control are sensory input, executive control, voluntary behavior, the AAIS, action schemata, and REF, and the automatized action level. The structure of the affective system (not shown) is mediated by the limbic system, a lateral circuit passing through the hypothalamus regulating internal and hormonal processes, the cingulate cortex, the hippocampus, and amygdala.

**Table 1 brainsci-09-00107-t001:** The rating scale in the Vividness of Visual Imagery Questionnaire [4].

Rating	The Image Aroused by an Item Might Be
1	Perfectly clear and as vivid as normal vision
2	Clear and reasonably vivid
3	Moderately clear and vivid
4	Vague and dim
5	No image at all, you only “know” that you are thinking of an object

**Table 2 brainsci-09-00107-t002:** Items in the Vividness of Visual Imagery Questionnaire [4] *.

Item	Theme	Description
	Relative or friend ^†^	For items 1 to 4, think of some relative or friend whom you frequently see (but who is not with you at present) and consider carefully the picture that comes before your mind’s eye.
1	Relative or friend	The exact contour of face, head, shoulders, and body.
2 *	Relative or friend	Characteristic poses of head, attitudes of body, etc.
3 *	Relative or friend	The precise carriage, length of step, etc. in walking.
4	Relative or friend	The different colors worn in some familiar clothes.
	Natural scene: Rising sun	Visualize a rising sun. Consider carefully the picture that comes before your mind’s eye.
5 *	Natural scene: Rising sun	The sun is rising above the horizon into a hazy sky.
6 *	Natural scene: Rising sun	The sky clears and surrounds the sun with blueness.
7 *	Natural scene: Rising sun	Clouds. A storm blows up, with flashes of lightening.
8 *	Natural scene: Rising sun	A rainbow appears.
	Shop	Think of the front of a shop which you often go to. Consider the picture that comes before your mind’s eye.
9	Shop	The overall appearance of the shop from the opposite side of the road.
10	Shop	A window display including colors, shape, and details of individual items for sale.
11	Shop	You are near the entrance. The color, shape, and details of the door.
12 *	Shop	You enter the shop and go to the counter. The counter assistant serves you. Money changes hands.
	Natural scene: Lake	Finally, think of a country scene which involves trees, mountains, and a lake. Consider the picture that comes before your mind’s eye.
13	Natural scene: Lake	The contours of the landscape.
14	Natural scene: Lake	The color and shape of the trees.
15	Natural scene: Lake	The color and shape of the lake.
16 *	Natural scene: Lake	A strong wind blows on the tree and on the lake causing waves.

* Eight of 16 items indicate activity or movement (marked *). † The first four items are from Peter Sheehan’s (1967) shortened form of the questionnaire designed by Betts (1909).

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
