# Peer review of "I Am Conscious, Therefore, I Am: Imagery, Affect, Action, and a General Theory of Behavior"

_brainsci, 2019, doi:10.3390/brainsci9050107_

Reviewer 1 Report

I found the paper 'I Am Conscious, Therefore I Am: Imagery, Affect, Action, And A General Theory Of Behaviour' to be basically scientifically sound, novel, of a good quality, and important to the related field of research. I recommend some minor issues.

Summary

The paper 'I Am Conscious, Therefore I Am: Imagery, Affect, Action, And A General Theory Of Behaviour' presents the argument that consciousness is striving for 'stability and equilibrium', and that imagery plays a central role in the achievement of equilibrium.

I find no problems with with the study’s method or analysis. However, there are several minor issues to be dealt with

Say earlier (e.g. in 'Prelimaries' section), and the make clear, using examples, how imagery relates to psychological homeostasis. We are told that ‘Consciousness is an open system having many relations to its mental, physical and social surroundings. Changes in these surroundings produce internal ‘disturbances’ of the system that require adjustment, adaptation or correction. ‘ This makes sense, but what role exactly does imagery play in ‘adjustment, adaptation or correction’?  Is it in making predictions?

To declare consciousness to be the 'central executive system' is problematic. There needs to be at least an acknowledgement of the growing consensus among students of the cognitive sciences that many of the contents of “consciousness” are formed ‘backstage’ by non-conscious systems - if only to be refuted. Is the argument that consciousness as personal awareness is the central executive system? Or does this lie in the contents of consciousness - beliefs, perceptions, etc? There needs to be an argument for why consciousness as personal awareness (if that is indeed how consciousness is being defined) should be considered the central executive sytem. For counter arguments, see Libet's classic experiment, but also Soon, C. S., Brass, M., Heinze, H.-J. & Haynes, J.-D. Nature Neurosci. doi: 10.1038/nn.2112 (2008) and, most extremely, 'Chasing the Rainbow: The Non-conscious Nature of Being' https://doi.org/10.3389/fpsyg.2017.01924

The central problem lies in aligning imagery vividness with both consciousness and cognitive processes, e.g. 'Imagery vividness is essential to imagining, remembering, thinking, predicting, planning, and acting'. Most people do not have very vivid imagery, as population studies with the VVIQ demonstrate – e.g. Bill Faw's (2009) Conflicting intuitions may be based on differing abilities - evidence from mental imaging research. Journal of Consciousness Studies, 16, 45-68. But the implication of the statement is that their cognitive processes depend on their imagery having vividness.

There is an emergent body of literature suggesting people without imagery still imagine, remember, and certainly ‘think’, and certainly ‘act' (see Zeman et al 2015). Would more vividness enhance cognitive processes? It may be more truthful to say that 'imagery plays a key role in most people’s imagining, remembering, etc'.

Similarly, the claim that ‘Vividness of mental imagery plays an essential role in the control of behaviour’ is problematic, because it implies that vividness, rather than, mental imagery per se, is essential. It implies that lesser vividness entails less control. However, it is the case that the 2-3% percent of the population (Faw 2009, Zeman et al 2015) who do not experience visual mental imagery are not less able to control their behaviour than those who can visualise.

Relatedly, is imagery vividness beneficial for homeostasis? Proportionally so? i.e. the more vivid, the better? The counter-position is that there are correlations of imagery strength with schizophrenia (Sack, A. T., van de Ven, V. G., Etschenberg, S., Schatz, D., and Linden, D. E. J. (2005). Enhanced vividness of mental imagery as a trait marker of schizophrenia? Schizophr. Bull. 31, 97–104. doi: 10.1093/schbul/sbi011), and PTSD (Holmes EA, Grey N, Young KA. Intrusive images and “hotspots” of trauma memories in posttraumatic stress disorder: an exploratory investigation of emotions and cognitive themes. J Behav Ther Exp Psychiatry (2005) 36(1):3–17. doi:10.1016/j.jbtep.2004.11.002) - that would seem to be undermine or represent a breakdown in homeostasis. So instead is there an 'ideal' imagery strength, i.e. present, but not too strong?

An inconsistency: on page 3, there is a ‘ceaseless progression of ideas and associations’ that occur without ‘visual signposting’; but on page 4: ‘thinking’ has an ‘essentially visual nature’. So by ‘thinking’ do we mean voluntary thought, specifically planning, etc? Or maybe truer to say mental imagery is a ‘commonly-used tool in planning, decision-making, recollecting, etc’.

The claim that ‘for a small minority [who do not experience visual mental imagery] consciousness is a pallid and abstract process’ is problematic and needs addressing. Having already equated consciousness with existence - ‘I am consciousness, therefore I am’ – the implication is that because these individuals have 'pallid' consciousnesses, and therefore a pallid selfhood. The emergent literature (Zeman et al 2015) suggests otherwise – that they lead full and often creative lives, living with a variant in cognitive processing, and that they report compensatory strengths elsewhere (in verbal, logical, domains), as you note in p13.

Lastly, it does not help to have a list of claims about consciousness 'which have a good chance of being true' on pp13-14. It might be easier to have something succinct like ‘consciousness is both experience and contents: personal awareness and contents of consciousness (beliefs, perceptions, etc – which might or might not enter personal awareness). The question then is what part of this is ‘the central executive system’?

Overall, I think the piece is scientifically sound and presents an intriguing new perspective and on imagery, and on imagery's role in individuals' relationships with their environments. However, counter-arguments regarding imagery's relative unimportance to consciousness, and consciousness's relative unimportance to decision-making, etc, need to be engaged with, in order to shore up the article's own argument.

Author Response

I thank the reviewer for the many helpful comments, which I address in the following response.

I found the paper 'I Am Conscious, Therefore I Am: Imagery, Affect, Action, And A General Theory Of Behaviour' to be basically scientifically sound, novel, of a good quality, and important to the related field of research. I recommend some minor issues.

Summary

The paper 'I Am Conscious, Therefore I Am: Imagery, Affect, Action, And A General Theory Of Behaviour' presents the argument that consciousness is striving for 'stability and equilibrium', and that imagery plays a central role in the achievement of equilibrium. 

I find no problems with with the study’s method or analysis. However, there are several minor issues to be dealt with

Say earlier (e.g. in 'Prelimaries' section), and the make clear, using examples, how imagery relates to psychological homeostasis. We are told that ‘Consciousness is an open system having many relations to its mental, physical and social surroundings. Changes in these surroundings produce internal ‘disturbances’ of the system that require adjustment, adaptation or correction. ‘ This makes sense, but what role exactly does imagery play in ‘adjustment, adaptation or correction’?  Is it in making predictions?

Thank you for this excellent point. I have added a paragraph at the end of the Preliminaries section to explain the role of imagery in ‘adjustment, adaptation or correction’ with examples.

To declare consciousness to be the 'central executive system' is problematic. There needs to be at least an acknowledgement of the growing consensus among students of the cognitive sciences that many of the contents of “consciousness” are formed ‘backstage’ by non-conscious systems - if only to be refuted. Is the argument that consciousness as personal awareness is the central executive system? Or does this lie in the contents of consciousness - beliefs, perceptions, etc? There needs to be an argument for why consciousness as personal awareness (if that is indeed how consciousness is being defined) should be considered the central executive system. For counter arguments, see Libet's classic experiment, but also Soon, C. S., Brass, M., Heinze, H.-J. & Haynes, J.-D. Nature Neurosci. doi: 10.1038/nn.2112 (2008) and, most extremely, 'Chasing the Rainbow: The Non-conscious Nature of Being' https://doi.org/10.3389/fpsyg.2017.01924

Thank you for mentioning this issue about non-conscious control. I agree 100%. I need to reinforce the idea that the central control system is hierarchical with an Executive, or Meta, Level, a Schema Level and an Automatized Level. It is the Executive Level that sets goals, monitors progress, and directs the Schema Level, which executes actions, and makes adjustments. The Meta Level exists within Consciousness while the majority of behaviour that is routine, falls within the Schema and Automatized Levels and does not require Consciousness. However conscious mental images are available at Schema Level as when required, e.g. in mental simulation, design and other creative performances. I provide more detail in an extra paragraph in section 12.

The central problem lies in aligning imagery vividness with both consciousness and cognitive processes, e.g. 'Imagery vividness is essential to imagining, remembering, thinking, predicting, planning, and acting'. Most people do not have very vivid imagery, as population studies with the VVIQ demonstrate – e.g. Bill Faw's (2009) Conflicting intuitions may be based on differing abilities - evidence from mental imaging research. Journal of Consciousness Studies, 16, 45-68. But the implication of the statement is that their cognitive processes depend on their imagery having vividness. 

Thank you for this point. I have changed the word ‘essential’ to ‘beneficial’, which is more accurate.

There is an emergent body of literature suggesting people without imagery still imagine, remember, and certainly ‘think’, and certainly ‘act' (see Zeman et al 2015). Would more vividness enhance cognitive processes? It may be more truthful to say that 'imagery plays a key role in most people’s imagining, remembering, etc'.

Thank you for this comment. I have inserted an extra paragraph at the end of section 6 concerning the evidence that a few people lack mental imagery (or the ability to voluntarily control it) and yet manage almost all skills and tasks.

Similarly, the claim that ‘Vividness of mental imagery plays an essential role in the control of behaviour’ is problematic, because it implies that vividness, rather than, mental imagery per se, is essential. It implies that lesser vividness entails less control.

I have deleted ‘Vividness of’ from the sentence quoted.

However, it is the case that the 2-3% percent of the population (Faw 2009, Zeman et al 2015) who do not experience visual mental imagery are not less able to control their behaviour than those who can visualise.

I have inserted an extra paragraph at the end of section 6 concerning the evidence that a minority of people lack mental imagery or lose mental imagery and yet manage almost all skills and tasks.

Relatedly, is imagery vividness beneficial for homeostasis? Proportionally so? i.e. the more vivid, the better? The counter-position is that there are correlations of imagery strength with schizophrenia (Sack, A. T., van de Ven, V. G., Etschenberg, S., Schatz, D., and Linden, D. E. J. (2005). Enhanced vividness of mental imagery as a trait marker of schizophrenia? Schizophr. Bull. 31, 97–104. doi: 10.1093/schbul/sbi011), and PTSD (Holmes EA, Grey N, Young KA. Intrusive images and “hotspots” of trauma memories in posttraumatic stress disorder: an exploratory investigation of emotions and cognitive themes. J Behav Ther Exp Psychiatry (2005) 36(1):3–17. doi:10.1016/j.jbtep.2004.11.002) - that would seem to be undermine or represent a breakdown in homeostasis. So instead is there an 'ideal' imagery strength, i.e. present, but not too strong?

I have added a brief discussion of vividness and homeostasis to section 12 (bottom of p. 20). I mention the intrusive imagery in psychiatric disorders in section 9 and give a reference [56]. I have added a few sentences on the subject of hallucinations and a reference which gives a definition of hallucinations [57]. A paper about the spectrum of conscious experience is in preparation and I have added a few words and a reference to this [58].

An inconsistency: on page 3, there is a ‘ceaseless progression of ideas and associations’ that occur without ‘visual signposting’; but on page 4: ‘thinking’ has an ‘essentially visual nature’. So by ‘thinking’ do we mean voluntary thought, specifically planning, etc? Or maybe truer to say mental imagery is a ‘commonly-used tool in planning, decision-making, recollecting, etc’.

Thank you for pointing out this inconsistency. The term ‘visual signposting’ has been deleted. I have left the word ‘thinking’ in place.

The claim that ‘for a small minority [who do not experience visual mental imagery] consciousness is a pallid and abstract process’ is problematic and needs addressing. Having already equated consciousness with existence - ‘I am consciousness, therefore I am’ – the implication is that because these individuals have 'pallid' consciousnesses, and therefore a pallid selfhood.

The emergent literature (Zeman et al 2015) suggests otherwise – that they lead full and often creative lives, living with a variant in cognitive processing, and that they report compensatory strengths elsewhere (in verbal, logical, domains), as you note in p13.

Thank you for this correction. I have removed ‘is a pallid and abstract process’ to avoid confusion. I have inserted the idea of compensatory strengths, which is also mentioned elsewhere in the paper.

Lastly, it does not help to have a list of claims about consciousness 'which have a good chance of being true' on pp13-14. It might be easier to have something succinct like ‘consciousness is both experience and contents: personal awareness and contents of consciousness (beliefs, perceptions, etc – which might or might not enter personal awareness). The question then is what part of this is ‘the central executive system’?

I have deleted the phrase “which have a good chance of being true’. However, I would prefer to leave the list of claims about Consciousness, as it is the core topic of the paper. I have indicated which items on the list are a part of the central executive system.

Overall, I think the piece is scientifically sound and presents an intriguing new perspective and on imagery, and on imagery's role in individuals' relationships with their environments. However, counter-arguments regarding imagery's relative unimportance to consciousness, and consciousness's relative unimportance to decision-making, etc, need to be engaged with, in order to shore up the article's own argument.

Thank you for your helpful and insightful comments. I have engaged with the counter-arguments and suggested why I think they are deficient.

I hope you find the revised version of the paper stronger as a consequence of the additions and amendments resulting from your, and the second reviewer’s, comments. 

Reviewer 2 Report

I like the paper and consider it relevant. The observations I make below aim at helping the author improve the paper, if he so desires.

The author rejects reductionism, saying, for instance, "a reductive approach is not required or desired and is not the approach taken here." He goes on to reject materialism (i.e. that behavior can be reduced to physico-chemical processes in the brain) and at times seems to reject evolution by natural selection. I think the paper would benefit from a more explicit stance in this regard. For instance, is the author saying that consciousness is an irreducible aspect of nature? If so, what makes the author think so?

It may also be too much to go against both material reductionism and evolution by natural selection in the same paper, without a lot more substantiation. For instance, the author writes:

>> Psychological homeostasis, as a process of Consciousness, is intentional, purposeful and driven by the desire for security, safety and equilibrium <<< p="">

Why couldn't this process not result from natural selection? Even if the author assumes that consciousness is not reducible to physiology, the particular behaviors it manifests (such as the need for homeostasis) can still be the result of natural selection. Or not?

It could be helpful if the author made clearer and more explicit, already early on, the connection between the fundamental, universal conscious desire for psychological homeostasis (your working principle) and the degree of vividness of inner imagery. In the abstract, for example, the connection is not made (at least not clearly). In other words, I miss a direct, explicit, early answer to the question: Why is high vividness required, or in some way relevant, for the achievement of psychological homeostasis? This is tackled later in the paper, but in the initial pages the reader is left to wonder.

The author argues that consciousness has a key role in the behavior of organisms. However, it's possible and coherent to conceive of UNconscious living beings that exhibit all the adaptive behaviors of conscious beings. Indeed, it is entirely conceivable and coherent to state that all structural and functional characteristics of living beings could be explained without them coming accompanied by conscious experience. It is metaphysically possible that they could all happen "in the dark" and be equally effective. This is the so-called "p-zombie" hypothesis in philosophy of mind.

There are also many who claim that unconscious mental processes can perform every function that apparently happens within consciousness. See e.g.:

https://journals.sagepub.com/doi/abs/10.1111/j.1745-6916.2006.00007.x

http://www.oxfordscholarship.com/view/10.1093/acprof:oso/9780195307696.001.0001/acprof-9780195307696

So it's tricky to attribute to consciousness a major explanatory role in the behavior of organisms, unless the author actually means META-consciousness. The word 'consciousness' is usually interpreted as phenomenal consciousness, what-it-is-likeness, qualia. But when an experience is re-represented in mind, then one has meta-consciousness, or conscious meta-cognition, or self-reflection. If the latter is what the author means by 'consciousness,' then it can certainly have a key role in behavior, __but not necessarily for the fact that it is also phenomenal__. Organisms could have access consciousness (a-consciousness) and be able to meta-cognize, but without there being anything it is like to be them.

Author Response

Thank you for your pertinent and constructive comments. My replies are listed below.

I like the paper and consider it relevant. The observations I make below aim at helping the author improve the paper, if he so desires.

The author rejects reductionism, saying, for instance, "a reductive approach is not required or desired and is not the approach taken here." He goes on to reject materialism (i.e. that behavior can be reduced to physico-chemical processes in the brain) and at times seems to reject evolution by natural selection. I think the paper would benefit from a more explicit stance in this regard. For instance, is the author saying that consciousness is an irreducible aspect of nature? If so, what makes the author think so?

Thank you for making these comments. I have amended the first two sentences of the Abstract and relevant sections of the paper to clarify my stance on evolution by natural selection. I do not reject the natural selection idea, but I wish to nuance it with ideas from evolutionary biology, e.g. the thought that human homeostatic activity in the form of niche construction generates much less variation in the source of selection than where there is no feedback from organisms’ activities to the environment. ideas from evolutionary biology (e.g. Laland et al., 2017).

It may also be too much to go against both material reductionism and evolution by natural selection in the same paper, without a lot more substantiation. For instance, the author writes:

>> Psychological homeostasis, as a process of Consciousness, is intentional, purposeful and driven by the desire for security, safety and equilibrium <<< span="">

Why couldn't this process not result from natural selection? Even if the author assumes that consciousness is not reducible to physiology, the particular behaviors it manifests (such as the need for homeostasis) can still be the result of natural selection. Or not?

Good point, thank you, I agree 100%, and I have attempted to clarify the relevant section of the paper concerning homeostasis and evolutionary selection. I have included the following on page 2: “Homeostatic striving for security, stability and equilibrium is a precondition for well-being. One major form of behavioural homeostasis is niche construction which alters ecological processes, modifies natural selection and contributes to inheritance through ecological legacies.”

It could be helpful if the author made clearer and more explicit, already early on, the connection between the fundamental, universal conscious desire for psychological homeostasis (your working principle) and the degree of vividness of inner imagery. In the abstract, for example, the connection is not made (at least not clearly). In other words, I miss a direct, explicit, early answer to the question: Why is high vividness required, or in some way relevant, for the achievement of psychological homeostasis? This is tackled later in the paper, but in the initial pages the reader is left to wonder.

This is a fundamental point, also made by the other reviewer. I have made this point as clearly as possible by amending the Abstract by adding: “The fundamental, universal conscious desire for psychological homeostasis benefits from the degree of vividness of inner imagery.”

The author argues that consciousness has a key role in the behavior of organisms. However, it's possible and coherent to conceive of UNconscious living beings that exhibit all the adaptive behaviors of conscious beings. Indeed, it is entirely conceivable and coherent to state that all structural and functional characteristics of living beings could be explained without them coming accompanied by conscious experience. It is metaphysically possible that they could all happen "in the dark" and be equally effective. This is the so-called "p-zombie" hypothesis in philosophy of mind.

The zombie is a fascinating hypothetical concept, but nobody has ever yet claimed to know one, be one or have built one. This is a complex issue and by raising it, which is a perfectly valid thing to do, you set the author what may appear to be a difficult challenge. Why did consciousness evolve?  It is a great question. I suggest the answer lies in the significant evolutionary advantages that accrue from Consciousness through the mechanism of homeostasis. I mention this possibility at several points in the paper. I am not alone in this view. J Scott Turner and Antonio Damasio have reached a similar conclusion. I have added a sentence or two at the end of section 12 (immediately above Figure 9) to make this position as clear as possible.

There are also many who claim that unconscious mental processes can perform every function that apparently happens within consciousness. See e.g.:

https://journals.sagepub.com/doi/abs/10.1111/j.1745-6916.2006.00007.x

This is a very relevant issue to raise. However, I do not agree with the claim that unconscious mental processes can perform every function that apparently happens within consciousness. I have seen no evidence to demonstrate that unconscious processes can entirely by themselves: reflect on past and future behaviour, imagine, image, hallucinate, simulate skilled activity, design, inventions, create new objects and performances, develop disciplines, art, music, philosophy, medicine, science, weigh consequences pro and con before acting, set purposes and goals for actions, perceive qualia, or know what it is to be like, or discuss and explain theories of Consciousness, to mention just a few.

The abstract of the paper by Dijksterhuis and Nordgren states: “We present a theory about human thought named the unconscious-thought theory (UTT). The theory is applicable to decision making, impression formation, attitude formation and change, problem solving, and creativity. It distinguishes between two modes of thought: unconscious and conscious. Unconscious thought and conscious thought have different characteristics, and these different characteristics make each mode preferable under different circumstances…” Setting aside methodological issues in coming to conclusions such as: “participants who could think unconsciously for 7 min made even better decisions than participants who could think unconsciously for only 2 min”, the tasks investigated by these investigators are artificial, e.g. the weighing of positive and negative attributes, and unconscious thinking is produced by distraction. According to the authors, “Conscious thought Is guided by expectancies and schemas”, “strategic thought processes are inherently hierarchical, whereas automatic processes are not”, and unconscious thinking does not perform a decision making task in the same way as conscious thinking. I see no substantial conflict between the theory in my article and the UTT. However, my theory does not address the issue of how decisions are made, but is concerned with the organisation of a control system of Consciousness capable of making a meaningful life that includes making decisions.

http://www.oxfordscholarship.com/view/10.1093/acprof:oso/9780195307696.001.0001/acprof-9780195307696

In The New Unconscious, Ran R. Hassin, James S. Uleman, and John A. Bargh argue that

“unconscious processes seem to be capable of doing many things that were thought to require intention, deliberation, and conscious awareness…These processes range from complex information processing, through goal pursuit and emotions, to cognitive control and self-regulation”.

I do not see any conflict between my theory that contains hierarchical levels of control, including a meta-level that sets conscious purposeful goals, a schema-level dealing with the execution of action, and a level for automatized action, and the unconscious processes in Hassin, Uleman and Bargh’s edited book.

So it's tricky to attribute to consciousness a major explanatory role in the behavior of organisms, unless the author actually means META-consciousness. The word 'consciousness' is usually interpreted as phenomenal consciousness, what-it-is-likeness, qualia. But when an experience is re-represented in mind, then one has meta-consciousness, or conscious meta-cognition, or self-reflection. If the latter is what the author means by 'consciousness,' then it can certainly have a key role in behavior, __but not necessarily for the fact that it is also phenomenal__. Organisms could have access consciousness (a-consciousness) and be able to meta-cognize, but without there being anything it is like to be them.

Thank you for raising these points. Amendments and additions have enabled further clarification of the hierarchy of levels, especially the meta-level concept. Additions and amendments have been made throughout the article with these points in mind. I hope that you find the revised version a clearer, stronger statement.